# Beyond Reality—Extending a Presentation Trainer with an Immersive VR Module

**DOI:** 10.3390/s19163457

**Published:** 2019-08-07

**Authors:** Jan Schneider, Gianluca Romano, Hendrik Drachsler

**Affiliations:** 1DIPF Leibniz Institute for Research and Information in Education, Rostocker Straße 6, D-60323 Frankfurt am Main, Germany; 2Goethe University Frankfurt, Theodor-W.-Adorno-Platz, D-60323 Frankfurt am Main, Germany

**Keywords:** sensor-based learning support, Multimodal Learning Analytics, public speaking

## Abstract

The development of multimodal sensor-based applications designed to support learners with the improvement of their skills is expensive since most of these applications are tailor-made and built from scratch. In this paper, we show how the Presentation Trainer (PT), a multimodal sensor-based application designed to support the development of public speaking skills, can be modularly extended with a Virtual Reality real-time feedback module (VR module), which makes usage of the PT more immersive and comprehensive. The described study consists of a formative evaluation and has two main objectives. Firstly, a technical objective is concerned with the feasibility of extending the PT with an immersive VR Module. Secondly, a user experience objective focuses on the level of satisfaction of interacting with the VR extended PT. To study these objectives, we conducted user tests with 20 participants. Results from our test show the feasibility of modularly extending existing multimodal sensor-based applications, and in terms of learning and user experience, results indicate a positive attitude of the participants towards using the application (PT+VR module).

## 1. Introduction

Usage of smart devices connected to the Internet is becoming mainstream. Therefore, having almost instant access to virtually all information in the world is also becoming normal. However, mere access to information does not suffice for one to acquire and develop any type of skills since the development of some skills requires practice while practice alone will only lead to a certain level of performance. The study of Bloom and Sosniak [1] revealed how elite performers had practiced intensively with dedicated mentors. Moreover, studies have also shown that only individuals who indulge in deliberate practice end up having a superior level of performance [2,3].

Having devoted human mentors always available who support learners with their deliberate practice is not a feasible nor affordable solution. Digital technologies, such as Intelligent Tutoring Systems (ITS), can be used to provide learners with different learning interventions such as scaffolding, feedback, etc., hence supporting them with the deliberate practice of their skills. An example is the classic LISP tutor that helps learners to become acquainted with the LISP programming language [4]. Traditional ITSs have shown positive effects supporting skills in well-formed topics and knowledge domains [5] such as mathematics [6] or programming languages [4,7]. Learners can practice this type of skills while directly interacting with the ITS through precise and unambiguous operations (e.g., keyboard strokes, button clicks).

There is a large set of skills, such as artisanal work, arts, or sports that require a certain type of coordination to be performed, and, thus, their practice cannot be assisted through the use of traditional ITSs. Technologies such as sensors are becoming increasingly available [8]. It is thus possible to unobtrusively capture the learner’s performance for a vast amount of learning scenarios and, hence, expand the set of skills that can be trained with the use of ITSs [9]. Direct input interactions such as button clicks are unambiguous and straightforward to interpret for a computer. In contrast, sensor data are context-dependent and challenging to interpret, therefore data from multiple modalities is usually needed to make correct interpretations and inferences.

Bridging the use of multimodal data with learning theories is a goal of the research field called Multimodal Learning Analytics (MMLA) [10]. MMLA applications can track and analyze multiple aspects of a learning activity such as the learners’ behavior, physiological state, and environment [11], and have shown their potential to support a vast number of learning scenarios [9,11,12]. These learning scenarios can include the deliberate practice of sports activities [13], playing musical instruments [14], executing health procedures [15,16], as well as some 21st-century skills such as job interviews [17], negotiation scenarios [18], collaboration [19,20] and public speaking [21,22,23].

The development of MMLA applications presents multiple challenges, starting from the collection of sensor data to the phase where the results of the analyzed data are presented as feedback to the learners. The feedback itself is composed of multiple dimensions, which can include the timing of the feedback (real-time or delayed), the modality of the feedback (visual, auditory, haptic), the complexity of the feedback, etc. [24]. MMLA applications combine these dimensions presenting feedback in a wide variety of forms. For example, the application described in [23] displays simple corrective real-time feedback regarding basic public speaking skills on a screen. In contrast, the applications presented in [22,25] that are also designed to support the development of public speaking skills provide more complex feedback which is presented offline.

The feedback of MMLA applications can go beyond computer screens. The use of Augmented Reality (AR) overlays on top of real objects has shown to be a useful mechanism to provide feedback for scenarios ranging from the training of calligraphy [26] to more critical domains such as the maintenance procedure of aircraft [27] and planning of surgical procedures [28]. For learning scenarios that are high risk and/or can raise the anxiety in learners, feedback can also be presented inside Virtual Reality (VR) simulations. This type of feedback is exemplified in the studies of [29,30] that focus on training for disaster and emergency management, and the studies of [15,16] that train surgical procedures.

Feedback can also be incorporated into different modalities such as audio as shown in the example of gait correction [31], or haptic as shown in the example of practicing snowboarding [13]. Even a combination of multiple feedback channels has already been explored. One example is the calligraphy tutor [26] which uses audio and haptic channels. Another example is the combination of visual and haptic feedback channels for catheter operation training [16].

The variety of ways to present feedback to learners is vast and finding one most suitable for the specific learning scenario is challenging. This challenge increases when considering that the development of MMLA applications is a costly endeavor. A reason for this high cost is that most of these learning applications are built from scratch and exclusively targeted to address their specific problem. The use of generic sensor, analysis and feedback modules to develop customizable MMLA applications seems suitable for reducing the development costs of these applications and drive the field of MMLA forward [32,33].

We argue that this customizable modular approach can accelerate the development of feedback module prototypes, and hence the investigation feedback mechanisms that are suitable for a specific learning scenario. To test the customizable approach of feedback modules for MMLA applications, in this study we explore how the Presentation Trainer (PT) [23], an MMLA application designed to support the development of basic skills for public speaking, can be extended with a VR real-time feedback module.

The presented study consists of a formative evaluation that has two main objectives. The first objective aims to validate the underlying technical concepts required to use a modular approach to extend the feedback mechanism of an existing MMLA application. Therefore, the feasibility of using the *LearningHub* [32] is evaluated to extend the existing PT with the VR module. The second objective of this study is to investigate the user experience of this new VR module in combination with a new post-practice histogram and video analysis feature of the PT.

## 2. Presentation Trainer

The Presentation Trainer (PT) is an application that allows learners to practice their presentations while receiving feedback regarding basic non-verbal communication aspects for public speaking. In the original version of the PT, while the learner practices the presentation, the PT displays on a computer screen her mirrored image. Based on the current learner’s performance, the PT displays a maximum of one feedback instruction at a given time. The feedback displayed concerns the learner’s *posture*, *use of gestures*, *voice volume*, *speaking time and use of pauses*, *phonetic pauses* (filler sounds), and *facial expressions* (smiles). After practicing the presentation, the PT provides learners with a self-reflection module where it displays the analysis of the data tracked during the presentation allowing learners to self-reflect about their performance [25]. For this study, we added two new features to the PT: a VR real-time feedback module and a post-practice histogram and video analysis.

### 2.1. Non-Verbal Communication Recognition and Mistakes Identification

To track the non-verbal communication of the learner, the PT uses the Microsoft Kinect V2 sensor together with its SDK. The PT uses the depth camera and SDK of the Kinect to capture the facial expression of the tracked learner and the 3D positions of the body joints of the tracked learner. Examples of these positions are ElbowLeft.Position.X, ElbowLeft.Position.Y, ElbowLeft.Position.Z, and ShoulderRight.Position.X. The microphone arrays and SDK of the Kinect are used for speech recognition and tracking the general ambient volume.

The PT uses the 3D positions of the body joints provided by the Kinect SDK to infer the current posture of the learner. It uses a rule-based mechanism to identify “posture mistakes” such as crossing arms, crossing legs, slouching, hands behind the back, hands being covered. The PT tracks the *use of gestures* first by creating a spatial representation of the learner’s arms, forearms and shoulder blades, based on the 3D position of the joints from hands, elbows, shoulders, and neck. It then calculates the angles between forearms, arms and shoulder blades. Next, it compares the angles from the current and previous frames to identify whether the learner performed a gesture. If the learner has been speaking for 6 s without performing any gesture, a “*use of gestures mistake*” is recorded.

The PT creates a 0.64 s buffer with the absolute volume values retrieved from the Kinect V2 SDK. It creates an average volume value from the buffer, and compares this value with the following thresholds that can be set up at runtime: *is_speaking, speaking_soft*, and *speaking_loud*. Speaking time starts when the average volume rises above the *is speaking* threshold, and finishes when the average volume is below the *is_speaking* threshold for longer than 0.25 s. At that moment a “Pause” starts. Whenever the speaking time is longer than 15 s, a “*use of pauses mistake*” starts being recorded. “*Speaking soft mistake*” starts being recorded when the average volume rises above the *is speaking* threshold and is below the *speaking soft* threshold. “*Speaking loud mistake*” starts being recorded when the average volume rises above the *speaking loud* threshold.

To identify the *Phonetic Pauses* or filler sounds the PT feeds the speech recognition module of the Kinect SDK with a set of predefined words such as *ahmm, ehhm, ohm, aaah*. Whenever one of these words is recognized by the speech recognition engine, a “*Phonetic Pause mistake*” is recorded.

The Kinect SDK allows identifying when learners display a happy facial expression, whenever a learner does not show a happy facial expression for a period longer than 25 s a *facial expression mistake* starts to be recorded.

### 2.2. Virtual Reality Real-Time Feedback Module

The act of public speaking almost invariably provokes an immediate anxiety response [34]. VR treatment sessions have shown to be effective in reducing public speaking anxiety in university students [35]. Moreover, to create more inclusive educational practices for public speaking the study in [36] suggest that VR based tasks should be integrated into educational practices for practicing presentations. Therefore, we consider that the extension of the VR real-time feedback module (VR Module) to the PT will enhance the PT by creating more comprehensive and immersive learning experiences.

The VR Module was developed for the Microsoft Hololens in Unity. It allows participants to practice their presentation and receive the real-time feedback generated by the PT while being situated inside of a 3D representation of a virtual classroom (See Figure 1). Learners can explore the classroom with the direction of their gaze. The feedback instructions displayed by the VR Module appear in real-time and follow the gaze of the learners. They appear and disappear based on the feedback rules of the PT. These feedback instructions consist of short textual instructions in combination with icons (see Figure 1), with the purpose to add some clarity to the instruction.

### 2.3. Post-Practice Histogram and Video Analysis

The original interface of the PT constantly displays the mirrored image of the learner, raising the level of self-awareness. This original support of self-awareness is not presented in the VR Module. To support self-awareness this new version of the PT automatically records a video of the practice session, which can be reviewed in the new Post-Practice histogram and video analysis module (See Figure 2). This module provides learners with three new options. First, it allows learners to review the videos of their practice sessions. Second, it allows learners to look at the development of their performance, based on the objective measurements (percentage of time spent on mistakes) tracked by the PT. Third, it allows learners to rate their performance because according to experts in public speaking ultimately only the presenter is the one who can assess if the presentation went as planned [37].

### 2.4. Presentation Trainer—Virtual Reality Module Communication

Communication between the PT and the VR Module happens with the help of the *LearningHub* and its libraries [32] (See Figure 3). The *LearningHub* is a system whose aim is to collect data from multiple sensor applications to generate a unified multimodal digital experience of a learning task. To create a unified multimodal digital experience, all the data produced by multiple sensor applications need to be collected and integrated. Sensor applications produce an enormous amount of data at different rates. For example, a sensor application such as the PT produces 30 frames per second, and each frame contains the 3D coordinates of 25 joints plus the audio level. The study in [32] shows that collecting *all data* in real-time does not scale revealing a scalability challenge. Therefore, the *LearningHub* first synchronizes the different sensor applications and collects their data through batch processing. Finally, it creates a recording of a Meaningful Learning Task (MLT). MLT recordings can later be inspected and analyzed in a generic manner [33].

Real-time feedback is one important learning intervention that can be performed by MMLA applications. To address the scalability challenge and support real-time feedback, the *LearningHub* allows the collection of *selected data* in real-time. *Selected data* can be composed of an outlier value from *all data* of the sensor frame, or the result from the interpretation of *all data* from the sensor frame as exemplified with the PT’s feedback. The LearningHub collects the *selected data* and broadcasts it to subscribed real-time feedback applications. The real-time communication feature of the *selected data* between the sensor applications, *LearningHub*, and feedback applications is performed through the use of the UDP protocol.

Communication between the PT and the VR module goes through the *LearningHub*. Where it is necessary to configure the PT as a sensor application and the VR module as a feedback application. Detailed procedure for the use of the LearningHub can be found on [38].

## 3. Method

### 3.1. Objectives of the Study

Design-based research is an iterative research methodology, which consists of designing, developing, and testing prototypical solutions [39]. This methodology is commonly used in the learning sciences to create and refine educational interventions. The study described in this paper is a formative evaluation, which consists of the first full iteration for researching the extension of the PT with the VR module. This first iteration has a technical feasibility and user experience objective.

Regarding technical feasibility, we aim to address one important issue concerning the development of MMLA applications. The development of this type of applications is expensive since they are usually tailor-made and developed from scratch. Therefore, it is difficult to spot an MMLA application that has already been used in formal learning scenarios, despite progress in the research field.

All MMLA applications share common challenges concerning their development. These challenges include the collection and integration of data, analysis of data, and presentation of useful information to learners. We argue that a configurable modular approach, in which MMLA applications can be configured using sharable modules, will contribute to their development, distribution and practical usage. In [32] authors suggested the use of a modular customized approach for the development of MMLA applications. Authors in that study revealed a scalability challenge by showing that the collection and integration of *all sensor data* in real-time do not scale for multiple sensor applications. Therefore, batch processing has to be used to create generic multimodal recordings. As a downfall, this batch approach inhibits generic MMLA applications to provide real-time feedback to learners. Previous studies have shown that for aspects that can be corrected immediately, real-time feedback is more effective than delayed feedback [40,41,42,43]. Therefore, this scalability challenge introduces an important limitation to the creation of generic MMLA applications. To address this limitation and scalability challenge, we want to explore the feasibility of sending in real-time only *selected data* to real-time feedback applications. In other words, testing the feasibility of transmitting in real-time the feedback generated by sensor applications to real-time feedback applications. Thus, advancing the exploration of the development of customizable modular MMLA applications. To test our first approach towards the customization of MMLA applications with different real-time feedback modules, we derived our first research question:

RQ1 (Technical Feasibility): Is it feasible to send *selected sensor data* in real-time to customize and extend MMLA applications with real-time feedback modules?

We used this research question to derive a more specific one for our particular use case:

RQ1a: Can we use the modular approach facilitated by the *LearningHub* to extend the PT with VR real-time feedback to provide learners with more comprehensive and immersive learning experiences?

This study presents the first iteration of a design-based research methodology studying the extension of the PT with the VR module. As a formative evaluation, for this iteration, it is important to explore how learners perceive the experience of practicing with this new setup. The study of this user experience will deliver a general impression of how learners interact with the application, as well as identifying some of its strengths and weaknesses. Using the VR module, the learner can practice in a more realistic environment while receiving feedback from the PT. Nevertheless, in contrast to the original version of the PT that shows a mirrored image of the learner, when using the VR module learners are not able to directly see their performance. Thus, we added the post-practice histogram and video analysis feature to the PT. The purpose of this added feature is to allow learners to revise their performance and to provide them with a more comprehensive learning environment. We derived the following research question to explore the new learning environment offered by the PT:

RQ2 (User Experience): How do learners perceive the experience of practicing their presentations using the VR module, and reviewing them with the post-practice histogram and video analysis feature?

### 3.2. Participants

To answer our research questions, we conducted user tests with 24 participants, twelve females and twelve males. The age of the participants ranged between 21 and 54 years with an average age of 31. All participants were professionals working at our institute, with a similar cultural background. We recruited them by personally asking for their willingness to participate in the study.

### 3.3. Procedure

Each user test session in this study was individual. The procedure of each session (see Figure 4) starts with a brief lecture about non-verbal communication for public speaking. This lecture intends to explain to participants how to interpret and adapt their behavior based on the feedback provided by the VR module. The procedure continues with a second brief lecture about the creation of an Elevator Pitch. An Elevator Pitch is a 30 to 120 s long speech where a speaker introduces themselves, summarizes in lay terms what they do and explains why it is important. Next, participants have five minutes to create their pitch.

The next stage of the test comprises three consecutive practice sessions. Each practice session consists of participants practicing their Elevator Pitch using the VR module of the PT and reviewing their performance afterward. Participants practice the Elevator Pitches while wearing the Hololens and standing between two to three meters in front of the Kinect (see Figure 5). The calibration process for the practice of the Elevator Pitches is as follows. Before the sessions, the experimenters adjusted the volume thresholds for the room where the user tests take place and the expected distance between the participants and the Kinect sensor. This calibration is sufficient for the analysis of the voice volume and speaking time performed by the PT. To calibrate the VR room according to the height and gaze direction of the participant, the participant has to wear the Hololens, stand in the right position facing the Kinect sensor, and finally has to start manually the VR module.

After practicing each of the Elevator Pitches, the participant takes off the Hololens, sits down in front of a computer screen, and looks at the Post-Practice histogram and video analysis module. This with the purpose to review her performance.

Once the three practice sessions (Elevator Pitch + Performance Review) are over, the participant fills in a user experience questionnaire.

### 3.4. Apparatus and Material

The tools used as interventions for this study are the VR Module of the PT running on a Microsoft Hololens, the PT including its new post-practice histogram and video analysis feature, and the *LearningHub* allowing the connection between both applications. The PT and the *LearningHub* run on the same Windows 10 PC with a 2.3 GHZ 2-core CPU and 16 GB of memory.

During each practice session, the PT generates log files that are used to measure the performance of the participants. The log files include all events captured during the practice sessions. This means all mistakes captured regarding the learner’s posture, use of gestures, voice volume, speaking time and use of pauses, phonetic pauses, and facial expressions.

A post-test questionnaire is used to query all participants about their experience of practicing their Elevator Pitches using the VR module of the PT and reviewing their performance. The questionnaire is based on the user experience questionnaire developed for the MMLA 2015 grand challenge [44] and consists of twelve Likert-type questions with a scale from one to ten, one question regarding the novelty of the tool, and five open questions regarding learning, improvements, and weak and strong aspects of the application (see Appendix A). From the twelve Likert-type questions, five inquire about learning and self-reflection, three about the realism of interacting with the system, two about the perceived usefulness of the system, and two about the enjoyment of using the system.

During the test sessions, experimenters take notes regarding the technical operation of the application and the participants’ performance.

## 4. Results

The user test allowed us to obtain some technical and practical results about the use of the PT and its VR module. Communication between the PT and the VR module can only happen when the *LearningHub*, PT, and VR module applications are all running on computers sharing the same network, the firewall of the network allows UDP communication and the ports used for communication are not blocked. To assure this during the tests, the PC running the PT and *LearningHub* hosted its network. We then connected the Hololens to this network. During the tests on two occasions, the Hololens was not able to connect to that network. We solved this issue by restarting the Hololens. With the Hololens and PC connected to the same network, the conducted test presented no further problems regarding the transmission of the feedback produced by the PT and broadcast to the VR module.

One technical limitation experienced during the tests involves the use of the “Bloom Gesture” (hold a hand, palm up, with fingertips together. Then open hand) while giving a presentation. This gesture is reserved for the Hololens operating system, and it sends all running applications to the background. On three occasions during the tests, participants used the “Bloom Gesture” by accident interrupting their practice session.

A second technical limitation of this study concerns the size and behavior of the VR audience used for the VR module. Originally, we planned to have an audience representative of a small lecture (30–50 members), with every member of the audience displaying some idle movements and specific behaviors depending on the feedback received from the PT. However, the processing power of the Hololens did not allow us to do this. Thus, to make the application usable, we had to reduce the number of members of the VR audience to five and keep their behavior static by removing idle movements and restricting the occurrence of animations.

The user tests allowed us to collect some data on the practical aspects of practicing presentations with the studied tools. In terms of objective measurements, the PT records into log files the timestamp when a mistake is detected and when a mistake is no longer recognized. These log files allowed us to identify the percentage of time that participants spend on making mistakes (pTM) while practicing their presentations. We used the percentage of time spend on mistakes because according to our criterion the influence of a mistake during a presentation depends on the percentage of time that the mistake is being displayed. For example, it is worse to speak too softly throughout the whole presentation than to speak too softly on several occasions for short periods that in total last a fraction of the presentation. Table 1 shows the mean pTM values for each of the detected mistakes. Since multiple mistakes can happen at the same time, the total pTM value can be larger than one.

As shown in Table 1, participants in general reduced their total pTM throughout the practice sessions, showing the biggest improvement in the use of pauses (33.7%), followed by the use of gestures (32.7%). The reduction of the total pTM from the first to the third session corresponds to 28.6%. We conducted a paired *t*-test to compare the total pTM from the first and the last practice session. Participants in the first session had a larger pTM (M = 1.296, SD = 0.590) than in the third practice session (M = 0.925, SD = 0.552); t (23) = 3.609, p > 0.002. This shows that the participants’ practice sessions had a positive impact on their objective performance.

In terms of user experience, 22 of the participants reported to never have used a similar application in the past. We divided the user experience into four dimensions: perceived learning, perceived usefulness, enjoyment and realism of the interaction. Table 2 displays the results from the Likert-type questions regarding user experience. The average score for all the items regarding the perceived learning experience is 79.9%. By looking at the results individually, it is possible to identify that in general terms participants found the tested system useful for practicing their skills. They rated the application to be slightly better than learning in the traditional classroom setting. Regarding the added post-practice histogram and video analysis, participants reported them to be very helpful to become aware of their performance and improve their skills.

The perceived usefulness of the system was tested through the items inquiring about the motivation to use the system again to prepare for future presentations and recommending the system to a friend. The average score of the perceived usefulness of the system is of 75.4%, indicating that in general terms participants felt motivated to use the PT again and would recommend it to a friend. The average score obtained in terms of enjoyment is 66.3%. Participants stated that the PT is fun to use, but they would not necessarily like to use it in their spare time. The average score of the realism of the interaction is 59.2%. To calculate this realism of the interaction, first, we reversed the scale of the invasiveness feeling as suggested in [45], so that the direction of the scores of this item aligns with the direction of the scores for the rest of the items where a high score equals a good rating.

We follow the suggestions in [44] to get an impression of the Quality of Experience (QoE) that participants had while practicing their Elevator Pitches with the PT based on the reported user experience. We argue that none of the dimensions examined (perceived learning, perceived usefulness, enjoyment and realism of interaction with the PT) directly affect each other. Therefore, to extract the QoE we just calculated their average. The resulting QoE value is 70.1% (see Figure 6).

Regarding the open-ended questions from the questionnaire, when participants were asked: “What, if anything, do you feel like you learned from using the application?”, the most frequently reported answers dealt with the use of pauses, reported by thirteen participants. The next most frequent answer involves the use of gestures, mentioned by four of the participants. These reports align with the objective measurements extracted from the log files of the practice sessions, showing that the area displaying the biggest improvement is the use of pauses, followed by the use of gestures. Two participants reported being surprised by what their gestures look like. Three participants commented on learning about the feeling of doing a presentation and the importance of practicing them.

Participants reported four different technical negative aspects of the application. On four different occasions, participants reported that the Hololens is a bit too heavy and uncomfortable to wear. In two occasions participants reported that the field of view of the Hololens is too small. The other reported negative aspects concern the “Bloom gesture” and the low framerate of the Hololens.

Regarding some negative user experience aspects in four occasions, participants reported that they did not like to be interrupted by the “Stop Speaking” instruction, in two occasions participants reported that they needed more time to explore the application, and one participant reported that the feedback icons displayed were too small.

Participants also reported the three most positive aspects of the application. Twelve participants commented on the insightfulness provided by the application in terms of self-reflection and becoming aware of their performance. In five occasions participants mentioned that the video analysis was particularly insightful in this regard. In ten occasions participants stated the helpfulness of the real-time feedback. Four times participants stated that feeling immersed in the classroom was positive. Some other positive aspects mentioned by participants on two occasions were that the application was fun to use, that it helped them to improve their skills and the importance of looking at their progress.

In terms of suggested improvements, ten participants suggested improving the audience by adding more members to the audience and/or making them move and react to the presentation. Three participants suggested different training setups depending on the type of presentation (academic, business, formal) and preparation phases e.g., preparation phase including some suggestions of gestures, pauses, that can be used, and practice phase with feedback. Other recurrent suggestions were the use of lighter-weight glasses, and the use of VR glasses instead of Hololens.

During the tests, experimenters observing the session also got the impression that some participants had some difficulties with the comfortability of the Hololens. Another observation from the experiments includes the personal impression that many of the participants during the practice sessions mostly focused on exploring the use of the application, rather than on performing well. Experimenters also noted that some participants while following the self-reflection phase were not sure of how to use the provided information to improve their skills, and commented that it would be good to see some examples of good practices.

## 5. Discussion

The user tests conducted in this study delivered answers to our technical feasibility and user experience research questions. RQ1 deals with investigating the feasibility and technical implications for customizing and extending MMLA applications with real-time feedback modules. In particular, for this study, RQ1a explored the feasibility of using the *LearningHub* to extend the PT with VR real-time feedback to provide a more comprehensive and immersive learning experience. The tests indicated a satisfactory answer to RQ1a, by showing that the feedback generated by the PT was communicated as *selected data* to the VR module, hence helping learners to adapt their behavior also in real-time while practicing a presentation in a VR scenario. This satisfactory answer shows the feasibility of using the modular approach of *LearningHub* for the customization of real-time feedback for MMLA applications, in contrast to the currently common practice of developing tailored-made MMLA applications with fixed feedback mechanisms. We hypothesize that this customizable approach towards MMLA applications has the potential to facilitate re-usage of existing modules, code, best practices, and techniques, and hence drive the development of the field forward.

The user tests also revealed some insights into how learners experience practicing their presentations with the PT and its new features (the VR module and the post-practice histogram and video analysis), thus providing answers to our RQ2. The overall QoE for interacting with the PT reported by the participants was relatively high, especially regarding the perceived learning and usefulness of the system, which we consider to be the most important aspects of the application.

Concerning learning, the studies in [23,46] show that the mere act of practicing a presentation several times does not lead to measurable improvements, in contrast to practicing a presentation while receiving feedback. The log files collected in our user tests allowed us to retrieve objective measurements indicating that participants got on average a 28.6% improvement from the first practice session to the last, indicating that the feedback of the PT helped participants to improve.

In terms of perceived learning, results from the quantitative aspect of the questionnaire indicate that participants felt they learned a lot while using the PT. Results from the open-ended questions show that participants particularly reported learning about the use of pauses and gestures, which were also the biggest areas of improvement showed by the objective measurements, with an improvement of 33.7% and 32.7% respectively.

Answers to the open-ended questions concerning the most positive aspects of the application show that participants appreciated the ability to practice a presentation while receiving real-time feedback and feeling immersed in a classroom, consequently revealing the importance of extending the PT with the VR-module. Moreover, answers to these questions point out the relevance of the tool concerning the development of self-awareness, which the post-practice histogram and video analysis help to obtain. Some participants explicitly pointed out they were surprised about the aspect of their gestures. Overall, results show how in terms of perceived learning the combination of the VR module and post-practice histogram seem to be very helpful.

In addition to the satisfactory learning aspects obtained in the study, results also show that in general terms learners feel motivated to use the PT and find it fun to use.

Results of this study also pointed out two main negative aspects of practicing with the current set-up of the PT + VR module, which provides valuable information for improving the system. The first reported aspect concerns the number of members in the audience and their static appearance. This aligns with the reported results about the realism of the interaction, which was the aspect of the QoE that received the lowest rating. Unfortunately, a completely 3D modeled environment with animated characters, like the one developed for the VR module, turned out to be too demanding for the processing power of the Hololens. We consider that this issue can be solved with two different strategies. The first one consists in the use of more powerful hardware. The second one is by substituting some computationally costly 3D objects with 2D objects. This second strategy can be sufficient because learners do not move around and explore the 3D VR environment while practicing a presentation. A possible implementable solution for this second strategy is the use of so-called HDRI maps for the classroom as the environment and the possibility to design 2D audience members that react to the learner’s performance.

The second main negative technical aspect shown in the results deals with wearing the Hololens. Some participants, especially the ones already wearing very big glasses, found the Hololens to be a bit uncomfortable and heavy. The use of more comfortable VR glasses might be a solution to these issues and could also help to increase the perceived feeling of immersion while practicing with the PT. However, this option can raise additional issues. Learners will not see their body including arms and hands, making it harder for them to correctly adapt to some of the feedback instructions, because of the diminished awareness regarding their own body. This can affect the usability of the whole application, therefore at some point, there must be a compromise between the feeling of immersion, usability, and technology.

Practicing presentations with the PT showed to be a unique experience for most of the participants in this study. It is important to take this novelty factor into account before deriving conclusions regarding the experience of using the PT. This indicates the main limitation of this study. Due to time restrictions, participants in the study only had three consecutive practice sessions with the PT. We did not test the long-term usage of the PT. We assume that the usage of the PT over a longer time will result in a slight decrease in the overall positive experience shown in this study.

## 6. Future Work and Conclusions

In this study, we evaluated the addition of a real-time VR feedback module, and a post-practice histogram and video analysis feature to the PT, a tool designed to support the practice of basic non-verbal communication skills for public speaking. Technical results from the study show the feasibility of adding generic real-time feedback modules to MMLA applications such as the PT with the help of the *LearningHub*. This, in turn, shows how the development of MMLA applications can follow a customizable generic approach that facilitates their development process. We argue that the development of MMLA applications is a current bottleneck in the field of MMLA. Therefore, by providing a solution to deal with this bottleneck, this research in turn also supports the distribution of best practices among the community and contributes to the advancement of the field. In line with this technical research, the plan is to conduct scalability studies with the *LearningHub*, where real-time feedback is broadcast from one MMLA application such as the PT to an N number of real-time feedback modules.

In terms of learning and user experience, the following main findings emerged.

Practicing with the PT helps learners to improve their performance.Learners appreciate practicing their presentations while receiving real-time feedback in the semi-immersive environment provided by the VR module.Learners found it insightful to become aware of their performance with the help of the post-practice histogram and video analysis feature.

Due to technical difficulties, the VR module turned out to be less immersive than expected. However, as a proof of concept, this study shows how VR real-time feedback modules, in combination with post-practice self-reflection tools that include video analysis, can create comprehensive, interactive, and immersive learning environments. For future work, firstly, it is important to test the broadcast of feedback of the PT to multiple feedback modules. An interesting setup for multiple feedback sources can be the addition of a haptic feedback module, since the addition of haptic feedback has already been shown to have a positive effect on the QoE [47]. A second important aspect to test for future work is the long-term effects of using the PT, which can be studied across a timespan of multiple months. Initial, mid, and final sessions can give an insight into the improvement and experience in the specific timespan.

Overall, the presented study provides an example of how the feedback of an existing MMLA application can modularly be customized with VR and/or AR technologies, creating a comprehensive learning environment for learners to acquire and develop their skills.

## Figures and Tables

**Figure 1 sensors-19-03457-f001:**
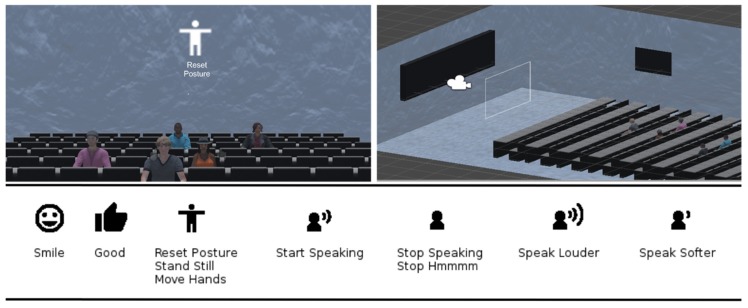
Top Left: User view while receiving Reset Posture instruction. Top Right: 3D Model of the classroom. Bottom: Feedback instructions used by the virtual reality (VR) module.

**Figure 2 sensors-19-03457-f002:**
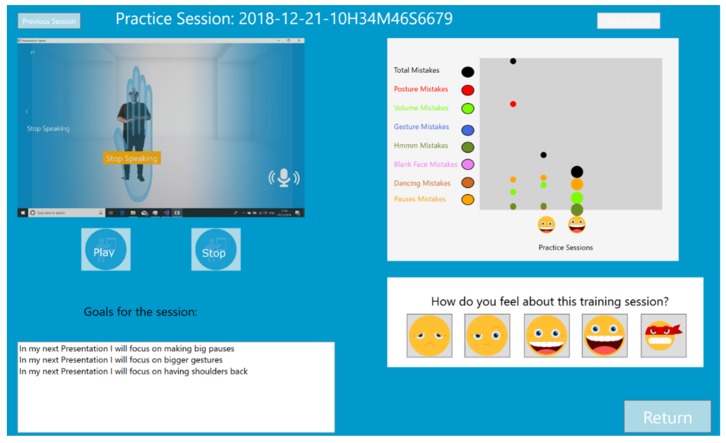
Post-practice Histogram. Top Left Video of the Practice Session. Top Right progress evolution according to percentage of time spent on mistakes. Bottom Left goals for the practice session. Bottom Right: self-evaluation.

**Figure 3 sensors-19-03457-f003:**
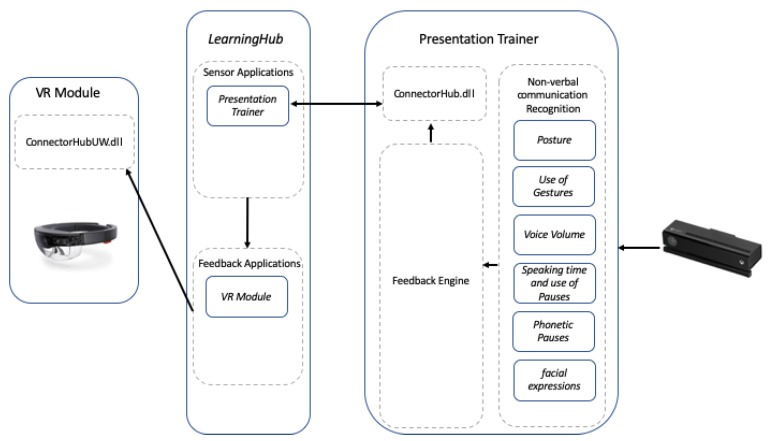
Block Diagram of the System Architecture: Presentation Trainer (PT), *LearningHub* and VR module.

**Figure 4 sensors-19-03457-f004:**
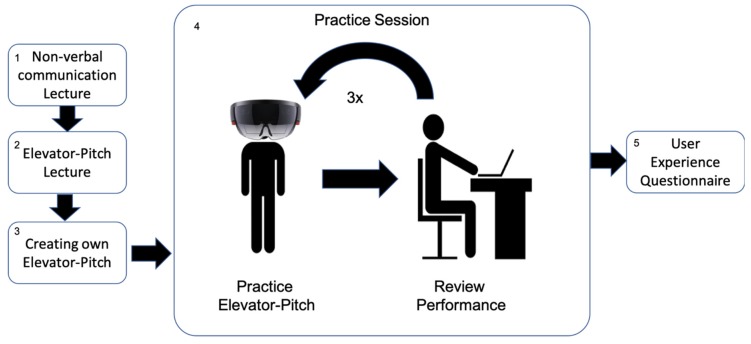
User test procedure.

**Figure 5 sensors-19-03457-f005:**
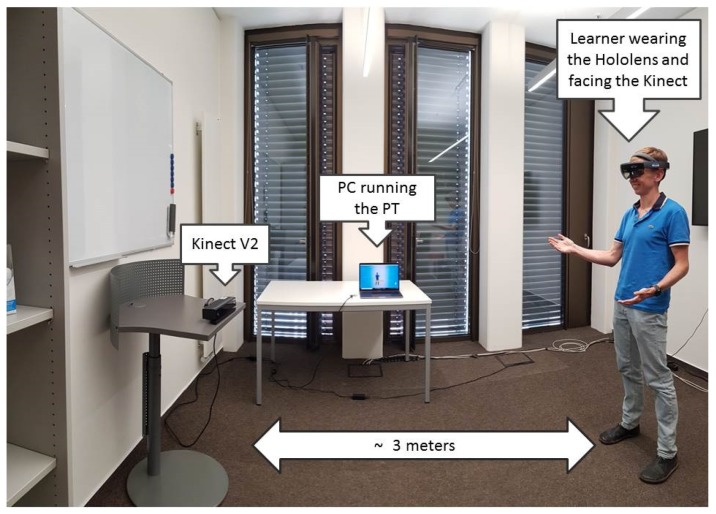
Picture of the experimental setup. Learner practicing an Elevator Pitch standing in front of the Kinect and Wearing the Hololens.

**Figure 6 sensors-19-03457-f006:**
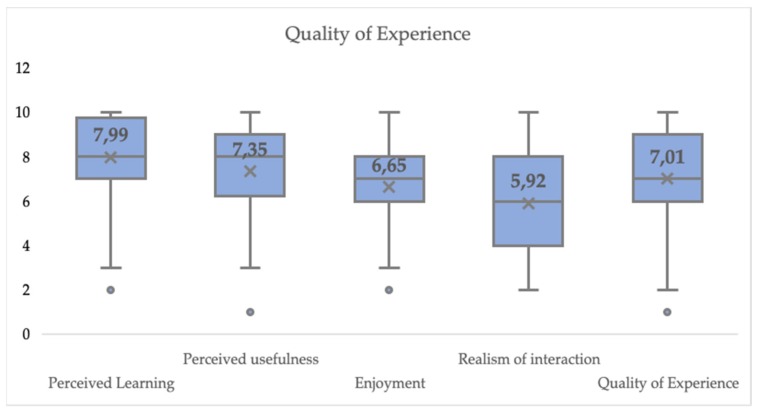
Quality of Experience for interacting with the PT + VR Module.

**Table 1 sensors-19-03457-t001:** Mean (M) of the pTM (0 to 1) of participants for the 3 practice sessions.

	Posture	Volume	Gesture	Phonetic Pauses	Serious Face	Dancing	Pauses	Total pTM
Session 1	0.142	0.242	0.297	0.008	0.049	0.027	0.531	1.296
Session 2	0.077	0.266	0.252	0.006	0.03	0.017	0.506	1.151
Session 3	0.091	0.221	0.200	0.009	0.02	0.021	0.352	0.925

**Table 2 sensors-19-03457-t002:** Reported results regarding perceived usefulness, enjoyment and interaction.

	Item	Mean	Standard Deviation
Learning	On a scale of 1 to 10, with 1 being not at all, and 10 being completely, do you feel like you learned anything while interacting with the application?	8.04	1.30
	On a scale of 1 to 10, with 1 being much worse, and 10 being much better, how does using this application compare to how you would normally learn the same content in a traditional classroom?	6.83	1.99
	Looking at the videos of me practicing made me aware of my performance (1 not at all–10 completely agree)	8.38	2.32
	Looking at the videos of me practicing helped me to improve my skills (1 not at all–10 completely agree)	8.09	1.64
	Looking at the development of my performance is helpful (1 not at all–10 totally agree)	8.63	1.01
Usefulness	On a scale from 1 to 10, with 1 being very bored, and 10 being very motivated, how motivated would you be to use this application again?	7.46	2.39
	I would recommend this application to a friend (1 not at all–10 totally agree)	7.63	1.97
Enjoyment	The application is fun to use (1 not at all–10 totally agree)	8	1.47
	On a scale from 1 to 10, with 1 being very likely, and 10 being very unlikely, how likely would you be to use this application in your free time?	5.25	2.51
Realism	I felt immersed in the scenario while practicing the presentations (1 not at all–10 totally agree)	5.75	2.49
	On a scale from 1 to 10, with 1 being very awkward, and 10 being very natural, how would you rate your experience with the application?	5.7	1.7
	On a scale from 1 to 10, with 1 being low, and 10 being very high, how invasive were the sensors being used to collect data about you?	4.7	2.38

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
