# Peer review of "Beyond Reality—Extending a Presentation Trainer with an Immersive VR Module"

_sensors, 2019, doi:10.3390/s19163457_

Round 1
Reviewer 1 Report
## Bibliography
Most of the references contains articles on how to develop speech skills in public. However, this article presents a complex system made by authors, containing hardware and software modules, based on various types of sensors. This system has to be described in the context of other similar systems developed in recent years.
There is very little bibliographic data on the use of VR technologies in training of any kind of activities, including real-time feedback. For some areas of training, these systems already exist on the market, not only in the research area (eg car simulators, planes ...). I suggest that these systems should be studied and evaluated comparatively in order to identify the following: what is the purpose to be pursued, what are the constructive and functional characteristics, how they are connected to human subjects, how to set and calibrate the systems, which is the way to generate feedback?
## Introduction
The original article [19] introduces the PT system in which it describes the way to improving speech abilities, namely pursuing nonverbal communication and correcting mistakes using two methods: tracking gestures and tracking the voice. For these actions it is use Kinect v2.
An observation: The equipment is mounted at a distance of 1.5 ... 3 m from the user so that the human silhouette can be spotted correctly. From this distance it becomes problematic to correctly capture the sound emitted by a human subject, the quality of the microphone being not very high.
There is no talk of installing, setting, calibrating the equipment for each user.
## Equipment. Methods
Missing a photo with the equipment
Missing a block diagram with the equipment running
In [19] only is talk about gesture tracking (angular changes between shoulder and arm, respectively arm and forearm) and voice. In the present article, line 88, there is also talk of facial expressions. With what sensors are they tracking? That is why I think there is a need for a more detailed explanation of the equipment and operating method.
There is no procedure to identify the angular changes of the skeleton components rebuilt by Kinect so that the movement / gesture mistake is sensed.
Feedback generated by the system is in the form of icons and small texts. Is it enough? Are tests done to see how long the subject gets used to them and does not care about them? It’s a haptic response necessary? It should be checked in the bibliography.
PT is a system designed and built using hardware and software libraries from the market and software modules developed by the team. A clear distinction must be made between them. I think we need to show how these hardware devices interconnect with the software, using block diagram, in which the data flow is clearly visualized: measurement, referral, decision making, command, warning, subject reaction ...
## Evaluation: case study
Being a system that contains equipment that comes in contact with the subject, I consider that the usage study should contain several chapters, with more questions: Engagement, Manipulability, Enjoyment, Realism, Usability, Overall experience.
Details of test questions can be found in the articles: (1) Hamam, A., & El Saddik, A.: Toward a mathematical model for quality of experience evaluation of haptic applications, IEEE Transactions on Instrumentation and Measurement, 62(12), 2013, pp. 3315-3322. (2) Hamam, A., Eid, M., & El Saddik, A.: Effect of kinesthetic and tactile haptic feedback on the quality of experience of edutainment applications. Multimedia tools and applications, 67(2), 2013, pp. 455-472. (3) Neupert, C., & Hatzfeld, C.: Evaluation of haptic systems, Engineering Haptic Devices, Springer London, 2014, pp. 503-524
I think test results can be much better visualized than graphics.
Author Response
Dear reviewer thank you very much for your supportive comments. We have considered each suggested comment thoroughly and are proud to present a significant improved paper for your attention.
Reviewer comment:
“Most of the references contains articles on how to develop speech skills in public. However, this article presents a complex system made by authors, containing hardware and software modules, based on various types of sensors. This system has to be described in the context of other similar systems developed in recent years.
There is very little bibliographic data on the use of VR technologies in training of any kind of activities, including real-time feedback. For some areas of training, these systems already exist on the market, not only in the research area (eg car simulators, planes ...). I suggest that these systems should be studied and evaluated comparatively in order to identify the following: what is the purpose to be pursued, what are the constructive and functional characteristics, how they are connected to human subjects, how to set and calibrate the systems, which is the way to generate feedback?”
Answer:
Indeed the previous version of the article mainly had bibliography concerning public speaking skills, for the revised version we expanded the introduction by adding key publications also from related fields , those can be found from line 48 to line 77. This new part of the introduction focuses on multiple ways to provide feedback and how feedback has been given by systems that have already been studied. We include literature from feedback provided in real-time vs. offline, AR, VR, multiple channels (visual, audio, haptic), and combined channels. We continued the introduction pointing out that there is no one fit all solution to provide feedback. Therefore, to move the field forward we suggest following a modular for the development of customizable MMLA applications, instead of the common tailor-made approach that is followed at the moment.
Reviewer Comment
““## Introduction
The original article [19] introduces the PT system in which it describes the way to improving speech abilities, namely pursuing nonverbal communication and correcting mistakes using two methods: tracking gestures and tracking the voice. For these actions it is use Kinect v2.
“In [19] only is talk about gesture tracking (angular changes between shoulder and arm, respectively arm and forearm) and voice. In the present article, line 88, there is also talk of facial expressions. With what sensors are they tracking? That is why I think there is a need for a more detailed explanation of the equipment and operating method.
There is no procedure to identify the angular changes of the skeleton components rebuilt by Kinect so that the movement / gesture mistake is sensed.”
“PT is a system designed and built using hardware and software libraries from the market and software modules developed by the team. A clear distinction must be made between them. I think we need to show how these hardware devices interconnect with the software, using block diagram, in which the data flow is clearly visualized: measurement, referral, decision making, command, warning, subject reaction …”
Answer
Thanks for the recommendation, indeed we agree that all this needs a better explanation, for that we added the subsection 2.1 to the article explaining how the mistakes are being recognized, what is performed by the SDK and what is performed by the PT. This section can be found from lines 109 to line 140. Furthermore, we also added a block diagram to better explain the data flow on page 6, Figure 3.
Reviewer comment:
An observation: The equipment is mounted at a distance of 1.5 ... 3 m from the user so that the human silhouette can be spotted correctly. From this distance it becomes problematic to correctly capture the sound emitted by a human subject, the quality of the microphone being not very high.
There is no talk of installing, setting, calibrating the equipment for each user.”
Answer:
Indeed this is a great suggestion therefore in In section 3.3 Procedure.
From lines 265 to 272 we added some text explaining the calibration process:
“Participants practice the Elevator-Pitches while wearing the Hololens and standing between 2m to 3m in front of the Kinect (see Figure 5). The calibration process for the practice of the Elevator-Pitches is as follows. Before the sessions, the experimenters adjusted the volume thresholds in accordance with the room where the user tests take place and the expected distance between the participants and the Kinect sensor. This calibration is sufficient for the analysis of the voice volume and speaking time performed by the PT. In order to calibrate the VR room according to the height and gaze direction of the participant, the participant has to wear the Hololens, stand in the right position facing the Kinect sensor, and finally has to start manually the VR module. ”
Reviewer comment:
## Equipment. Methods
Missing a photo with the equipment
Missing a block diagram with the equipment running
In section 3.3 Procedure we added a picture (Figure 5.) of the set up used for the user tests showing the equipment and how it was used
In section 2.4 Presentation Trainer - Virtual Reality Module communication we added a picture showing the Block diagram of all the components of the system
Reviewer comment:
“Feedback generated by the system is in the form of icons and small texts. Is it enough? Are tests done to see how long the subject gets used to them and does not care about them? It’s a haptic response necessary? It should be checked in the bibliography.”
Answer:
This is a great comment, and the questions are very interesting. Unfortunately, answering them is currently outside the scope of the study. However, in the future work section (lines 503 to 405) we already indicated, that we want to further explore this area of research.
“An interesting setup for multiple feedback sources can be the addition of a haptic feedback module since the addition of haptic feedback has already shown to have a positive effect on the QoE [42]. ”
Reviewer comment:
## Evaluation: case study
Being a system that contains equipment that comes in contact with the subject, I consider that the usage study should contain several chapters, with more questions: Engagement, Manipulability, Enjoyment, Realism, Usability, Overall experience.
Details of test questions can be found in the articles: (1) Hamam, A., & El Saddik, A.: Toward a mathematical model for quality of experience evaluation of haptic applications, IEEE Transactions on Instrumentation and Measurement, 62(12), 2013, pp. 3315-3322. (2) Hamam, A., Eid, M., & El Saddik, A.: Effect of kinesthetic and tactile haptic feedback on the quality of experience of edutainment applications. Multimedia tools and applications, 67(2), 2013, pp. 455-472. (3) Neupert, C., & Hatzfeld, C.: Evaluation of haptic systems, Engineering Haptic Devices, Springer London, 2014, pp. 503-524
Answer:
This was another great suggestion. Instead of using Engagement, Manipulability, Enjoyment, Realism, and Usability. We used Perceived Learning, Perceived usefulness, Enjoyment, and Realism because we found these dimensions to align better to the used questionnaire, our study setup, and the use of the PT in general. However, making these distinctions allowed us to get a better impression of the Quality of Experience of using the PT, pointing out Realism as the weakest part of it, which in turn aligned with the free comments reported by the participants.
In order to do this, we adjusted the following sections:
3.4 Apparatus and Material
Lines 296 - 298
“From the twelve Likert-type questions, five inquire about learning and self-reflection, three about the realism of interacting with the system, two about the perceived usefulness of the system, and two about the enjoyment of using the system.”
4 Results
Lines 347- 370
Figure 6.
5. Discussion
Lines 430 - 432
“The overall QoE for interacting with the PT reported by the participants was relatively high, especially regarding the perceived learning and usefulness of the system, which we consider to be the most important aspects of the application. ”
6. Future Work and Conclusions
Lines 503-505
“. An interesting setup for multiple feedback sources can be the addition of a haptic feedback module since the addition of haptic feedback has already shown to have a positive effect on the QoE [42].”
Reviewer 2 Report
The paper presents the combination of two systems to provide a virtual learning environment with analytical feedback related with multimodal sensors information.
The application is original and it covers a vibrant new area of research and application though it lacks of strictness. Please find some comments in the following:
· At section 2 the authors say: “Presentation Trainer – Virtual Reality Module communication” Technical acpects of the implementation should not be included at least they are an important part of the proposed idea or main objective of the paper. The sentence “In the case of the VR module, one needs to add a reference to the ConnectorHubUW.dll to the project, create and initialize a ConnectorHubUW.feedback object, and create a feedbackReceivedEvent event handler in order to catch the feedback sent from the LearningHub.” This paragraphs doesn’t provide any information to the reader to better understand the paper but it explains implementation details that could be incorporated in a GitHub readme file of the code used. The same applied to Figure 3.
· At line 152 the authors say that the first research question is : “RQ1 (Technical Feasibility): Can we use a modular approach to extend existing MMLA applications?” This is not a research question but a methodology for software development.
· The same applies to the second onw: “How do learners perceive the experience of practicing their presentations using the VR module, and reviewing them with the post-practice histogram and video analysis feature?” The users always will learn more doing a learning process, either virtual o physical, than not doing any at all.
· The authors use th “LearningHub” tool which is not explained in the paper. It will be useful to include the type of multimodal analysis that is done and describe the characteristics of the different type of media under analysis (e.g. how pauses or tone are obtained). This was published in other paper but a brief description will be useful referencing it as the authors did.
· Line 215-224 these sentences explain very basic communication network knowledge so it should be omitted in this type of paper. A brief subsection describing the whole architecture that has been used would me more useful that describing some parts of it.
· Lines 225-229 doesn’t provide relevant information for the reader. Any practical problem related with the used prototype should be solved before doing the study or it should be done again after they are fixed.
· Lines 230-236 This lines doesn’t provide insightful information. It would be interesting to see how the differences in the audience affects to the learning process (e.g. if the speaker has to pay attention to the audience will do the learning process slower? This could be a good research question for instance.)
· Lines 279-281. There is a lack of interpretation of the results, it just repeats the same information that is included in the table. There should be an insightful interpretation according to the domain of application. Overall, the results section lacks of this appreciation.
· Lines 315-319. There is not a possible relation between a software constraint and a research question. The application can be on real time or not but there is not any research question in that sentence. It is just a design property.
· Section 4 and 5 should be combined, section 5 is supposed to present a discussion on the results but it mostly present technical issues and problems that has been experienced (1 page is too much and it should be reduced to a couple of paragraphs and in the technical design not in the discussion section).
· In the presented study, there is not a control group to compare with. The author’s only show that using the system the user improves their performance but, using other “classical” learning techniques; wouldn’t the users also had improve their performance? There should be a control group to show this difference if it exists.
Overall the paper lacks of a clear definition of the study and conclusions that the authors want to achieve. Instead, the authors present an application (that was partially developed and presented in other paper) and try to proof two research questions that are not well presented. The first is not a research question but a software property and the second it not properly proof and the obtained results were expected (is expected that doing a learning process the user improve its performance) and not compare with other methodologies (the results should be compared with other learning methodologies that are not virtual). There should be a real research question to be answer in the study.
Author Response
Dear reviewer thanks for taking time into reviewing the paper and providing insightful comments. In general they help us to revise the whole paper and create an improved version of it. Next you can find some specific answers to your comments together with some concise modifications to the paper.
Reviewer comment:
“ At section 2 the authors say: “Presentation Trainer – Virtual Reality Module communication” Technical acpects of the implementation should not be included at least they are an important part of the proposed idea or main objective of the paper. The sentence “In the case of the VR module, one needs to add a reference to the ConnectorHubUW.dll to the project, create and initialize a ConnectorHubUW.feedback object, and create a feedbackReceivedEvent event handler in order to catch the feedback sent from the LearningHub.” This paragraphs doesn’t provide any information to the reader to better understand the paper but it explains implementation details that could be incorporated in a GitHub readme file of the code used. The same applied to Figure 3.”
“The authors use th “LearningHub” tool which is not explained in the paper. It will be useful to include the type of multimodal analysis that is done and describe the characteristics of the different type of media under analysis (e.g. how pauses or tone are obtained). This was published in other paper but a brief description will be useful referencing it as the authors did.”
Answer:
Thanks for this good recommendation. We agree the implementation and configuration details should be on the readme file of the Github link and not in the article. We will work on this for future releases of the PT. Furthermore, we removed these details from the article but expanded the description of the and functions of the LearningHub. Changes can be found in section 2.4 Presentation Trainer - Virtual Reality Module communication.
Lines 174 - 191
Reviewer comment:
“ At line 152 the authors say that the first research question is : “RQ1 (Technical Feasibility): Can we use a modular approach to extend existing MMLA applications?” This is not a research question but a methodology for software development.”
“Lines 315-319. There is not a possible relation between a software constraint and a research question. The application can be on real time or not but there is not any research question in that sentence. It is just a design property.”
Answer:
Many thanks for pointing out that this research question can be misunderstood. In order to improve the research question we took various steps: First, we further describe the state of the art of the Learning Hub in section 2.4, Second, On this basis we made the challenge of scalability of real-time data communication explicit.
Third, we adjusted the section 3.1 Objectives of the study. Lines 196 - 228. There we first state that the overarching methodology for the research is a design-based research methodology, second we state that the described study is reports on the first iteration of the methodology. There we argue that for this first iteration it is important to explore the technical feasibility of the study and the user experience.
We also rephrased the research questions:
“RQ1 RQ1 (Technical Feasibility): Is it feasible to send selected sensor data in real-time in order to customize and extend MMLA applications with real-time feedback modules?
We used this research question to derive a more specific one for our particular use case:
RQ1a: Can we use the modular approach facilitated by the LearningHub to extend the PT with VR real-time feedback in order to provide learners with more comprehensive and immersive learning experiences?”
Reviewer comment:
The same applies to the second onw: “How do learners perceive the experience of practicing their presentations using the VR module, and reviewing them with the post-practice histogram and video analysis feature?” The users always will learn more doing a learning process, either virtual o physical, than not doing any at all.
Answer:
Thank you for your feedback for improving research question 2. We modify the paper (section 3.1 Objectives of the study) to explicitly state the relevance of exploring the user experience of the learner. (Line 197-201; Line 230 - 233)
Regarding the usability aspect of RQ2, we consider that the way learners experience their interaction with the application will determine the amount of usage of the application, amount of practice of the learners, and therefore how much they learn at the end. If the experience of interacting with it is confusing, boring, or useless, they will never use the application to improve their learning . That is the reason why we consider it important to first do a formative study to explore the user experience before conducting an advanced A-B test experiment to compare the application with traditional learning scenarios. We consider this part of a Design-Based Research methodology that many similar researchers in the field apply for maturing their prototypes.
In terms of users always learning more by doing something than by not doing anything at al, Indeed we also think that users will learn more by doing something than by not doing anything at all. However, as stated in the introduction of the paper, just practicing without any guidance or feedback is not really effective: “The study of Bloom and Sosniak [1] revealed how all elite performers from the study had practiced intensively with devoted mentors. Moreover, studies have also shown that only individuals who indulge in deliberate practice end up having a superior performance [2,3].”
Moreover, the studies by (Barmaki & Hughes, 2015) and (Schneider, Boerner, van Rosmalen, Specht 2016) showed that in the case of public speaking real-time feedback is necessary for learners to improve certain non-verbal communication aspects. Practice without feedback showed no effect in the correction of these aspects. We addressed this point in our manuscript in section 5 Discussion lines 426-428.
Barmaki, R., & Hughes, C. E. (2015, November). Providing real-time feedback for student teachers in a virtual rehearsal environment. In Proceedings of the 2015 ACM on International Conference on Multimodal Interaction, 2015, pp. 531-537. ACM.
Schneider, J., Börner, D., van Rosmalen, P., Specht, M. Can you help me with my pitch? studying a tool for real-time automated feedback. IEEE Trans. Learn. Technol. 9, 2016, pp. 318–327.
Reviewer comment:
Line 215-224 these sentences explain very basic communication network knowledge so it should be omitted in this type of paper. A brief subsection describing the whole architecture that has been used would me more useful that describing some parts of it.
Answer:
We agree that it can be basic information however we still consider our feasibility research question to be very important. This paragraph for technical people might give no extra value, nonetheless, it allows us to answer our question. Furthermore, we expanded section 2.3 and added Figure 3. to explain the architecture used.
Reviewer comment:
Lines 225-229 doesn’t provide relevant information for the reader. Any practical problem related with the used prototype should be solved before doing the study or it should be done again after they are fixed.
Lines 230-236 This lines doesn’t provide insightful information.
Answer:
We consider that one of the main purposes for the type of formative study described in this paper is precisely to identify this type of mistakes in real scenarios with small scale studies, before doing a summative evaluation of the system.
Reviewers comment:
It would be interesting to see how the differences in the audience affects to the learning process (e.g. if the speaker has to pay attention to the audience will do the learning process slower? This could be a good research question for instance.)
Answer:
We completely agree that these are very interesting points to research, which can be researched in future iterations of the design-based research methodology, once some of the technical problems identified in this study are solved. The identification of these problems is part of the contribution of this study.
Reviewers comments:
Lines 279-281. There is a lack of interpretation of the results, it just repeats the same information that is included in the table. There should be an insightful interpretation according to the domain of application. Overall, the results section lacks of this appreciation.
Section 4 and 5 should be combined, section 5 is supposed to present a discussion on the results but it mostly present technical issues and problems that has been experienced (1 page is too much and it should be reduced to a couple of paragraphs and in the technical design not in the discussion section).
Answer:
Thanks for the recommendation, for the corrected version of the manuscript we added more interpretations to the section 4. Results (Lines 303 - 412). However, we tried to keep the interpretation of the results just in terms of the data obtained through the user tests. Therefore, we maintained and expanded section 5 Discussion (Lines 414-477) where we interpret the results based on our research questions, previous studies, broader implications of the results and also mentioned limitations of the study following the guidelines of the journal.
Reviewer comment:
In the presented study, there is not a control group to compare with. The author’s only show that using the system the user improves their performance but, using other “classical” learning techniques; wouldn’t the users also had improve their performance? There should be a control group to show this difference if it exists.
Answer:
Indeed testing the current setup against classical learning techniques is an interesting point to research. However, in the current cycle of the design-based research methodology the study presented in the paper consists of a formative study, testing the effectiveness of different interventions including the current setup of the PT was out of the scope of the study. However, while this is an interesting point that definitely is worth exploring, previous studies have shown that learners do not automatically correct their nonverbal communication by just practicing. Feedback is needed in order to correct some mistakes. The presented study shows already an improvement in the participants performance, showing that the feedback provided by the system was to certain degree effective. We pointed this out in the new version of the paper. Section 5 Discussion lines 433-435.
“Concerning learning, the studies in [23,41] show that the mere act of practicing a presentation several times does not lead to measurable improvements, in contrast to practicing a presentation while receiving feedback. ”
Reviewer 3 Report
it isn't clear if the questionnaire MMLA was applied to ¿how many participants?. I understood the idea is evaluate the user experience integrating VR Module, but it isn't clear.
In Figure 2, the emoticons with facial expressions related with scale 1 to 5, is supported by some author?
Author Response
Dear reviewer thank you very much for your supportive comments. We have considered each suggested comment thoroughly and are proud to present a significant improved paper for your attention.
Reviewer comment:
it isn't clear if the questionnaire MMLA was applied to ¿how many participants?. I understood the idea is evaluate the user experience integrating VR Module, but it isn't clear.
Answer
Indeed the questionnaire was applied to all participants. Originally we had 20 participants that we considered a good number for a formative study like this. However, we followed the recommendation of one of the reviewers and we added 4 participants more so we had a total of 24 participants at the end. To make it more explicit we added the section 3.2 Participants and in section 3.4 Apparatus and Material lines 291-292 we added the following:
“A post-test questionnaire is used to inquire all participants about their experience of practicing their Elevator-Pitches using the VR module of the PT and reviewing their performance.”
Reviewer comment:
In Figure 2, the emoticons with facial expressions related with scale 1 to 5, is supported by some author?
Answer
The precise use of emoticons to self assess the performance as far as we know are not related to a specific study, However, the action of self-assessing the practiced presentation is based on the study of:
Schneider, J., Börner, D., Van Rosmalen, P., & Specht, M. Presentation Trainer: what experts and computers can tell about your nonverbal communication. Journal of computer assisted learning, 33(2), 2017, 164-177.
Were experts interviewed in the study stated that ultimately only the presenter can assess if the presentation went as planned. We made this explicit in section 2.3 Post-Practice histogram and video analysis line 165, 166.
Reviewer 4 Report
The authors may want to include some papers using Microsoft Kinect with an immersive VR module in a similar context, such as 10.1145/2542050.2542060 and 10.1109/THMS.2015.2453203 (DOI).
The number of participants is quite small. If possible, please include more participants, especially females.
The quality of Figure 3 is quite low, and the information it provides was not described clearly.
Statistical testing methods should be included when discussing results.
A major revision is needed for this paper.
Author Response
Dear reviewer thank you very much for your supportive comments. We have considered each suggested comment thoroughly and are proud to present a significant improved paper for your attention.
Reviewer comment:
The number of participants is quite small. If possible, please include more participants, especially females.
Answer:
Thanks for the suggestion, in this short time we were able to include 4 female participants to the study. The results were almost no affected. One noticeable difference is that the difference between the measured performance from the 1st session in comparison to the 3rd session became more significant. Making the results and value of the application stronger.
Reviewer comment:
The quality of Figure 3 is quite low, and the information it provides was not described clearly.
Answer:
Another reviewer recommended skipping the technical details on how to connect the LearningHub and the PT, put this details in the github readme file of the LearningHub, and in the paper discussed better the properties of the system. We followed this suggestion and deleted that Figure 3. And re-wrote the whole section 2.4 Presentation Trainer - Virtual Reality Module communication, and added a new Figure showing the block diagram of the whole set up.
Reviewer comment:
The authors may want to include some papers using Microsoft Kinect with an immersive VR module in a similar context, such as 10.1145/2542050.2542060 and 10.1109/THMS.2015.2453203 (DOI).
Answer:
Those two papers were great suggestions and indeed are very relevant to this study. We were able to include one of them to the introduction of the paper that now included a much more extensive review of the state-of-the-art (Line 48 to Line 77).
Reviewers comment:
Statistical testing methods should be included when discussing results.
Answer:
We mostly used descriptive statistics for this study, we only used a T-test to compare the performances from 1st sessions with the performances of the 3rd sessions. Therefore, there were not many statistical methods to discuss. However, we improved the section 5 Discussion by comparing results with results from similar studies (Line 433 to Line 438). Moreover, we provided a more explicit including a numeric description to the discussion of some of the results (Line 439 to 442)
Round 2
Reviewer 1 Report
Line 114: Ref. facial expression. How can you track both the skeleton and the facial expression with one Kinect device? Do you have an own application software able to split the data from Kinect? The range between Kinect and the user is the same for both applications or you have to move the device?
Line 119: Ref. calculated angles. Angles that are visualized on Kinect's SDK interface are 2D format. In reality, the angles between the human skeleton segments are in a space coordinate system, so 3D. What Kinect sees are only projections on a plane of spatial angles. How did you solve this problem? Even if you have imposed angular values, the differences between 2D projections and 3D real angles may be significant, especially when the user rotates around the vertical axis.
Lines 115: Ref. Posture mistakes. Same issues as above. When rotating the user around the vertical axis, the skeleton segments overlap, giving rise to the wrong positions. How did you solve the problem?
Figure 5 - It's not very explicative. It can be improved.
Author Response
Dear Reviewer thank you so much for taking the time to review our paper one more time. We hope that we can clarify all of your comments now.
Comment:
“Line 114: Ref. facial expression. How can you track both the skeleton and the facial expression with one Kinect device? Do you have an own application software able to split the data from Kinect? The range between Kinect and the user is the same for both applications or you have to move the device?”
Answer:
We are using only one Kinect sensor V2 to capture both the facial expression and the body of the user. To make this clearer in the new version of the paper we added the following:
Lines: 118- 119:
"The PT uses the depth camera and SDK of the Kinect to get the facial expression of the tracked learner and the 3D positions of the body joints of the tracked learner."
Comment:
“Line 119: Ref. calculated angles. Angles that are visualized on Kinect's SDK interface are 2D format. In reality, the angles between the human skeleton segments are in a space coordinate system, so 3D. What Kinect sees are only projections on a plane of spatial angles. How did you solve this problem? Even if you have imposed angular values, the differences between 2D projections and 3D real angles may be significant, especially when the user rotates around the vertical axis.
Lines 115: Ref. Posture mistakes. Same issues as above. When rotating the user around the vertical axis, the skeleton segments overlap, giving rise to the wrong positions. How did you solve the problem?”
Answer:
The Kinect actually has a depth camera that can look in 3 dimensions. The Kinect SDK provides different values and attributes. Indeed, one can use the SDK of the Kinect to extract a 2D skeletal representation of the learner. It also provides a 3D coordinate for the joints of the tracked body, each joint has an X, a Y, and a Z value. For the analysis of the gestures and posture of the user, the PT does not use the 2D skeletal representation, it only uses the 3D values. Therefore, it never uses 2D projections.
To make this clear for readers of the paper, we modified the following lines:
Line 118 - Line 124:
"The PT uses the depth camera and SDK of the Kinect to get the facial expression of the tracked learner and the 3D positions of the body joints of the tracked learner. Examples of these positions are ElbowLeft.Position.X, ElbowLeft.Position.Y, ElbowLeft.Position.Z, and ShoulderRight.Position.X. The microphone arrays and SDK of the Kinect are used for speech recognition and tracking the general ambient volume.
The PT uses the 3D positions of the body joints provided by the Kinect SDK to infer the current posture of the learner."
Comment:
Figure 5 - It's not very explicative. It can be improved.
Answer:
Indeed, thanks for the suggestion. We changed figure 5 adding a figure that we consider to be more explicative.
Reviewer 2 Report
Please find attached my review including previous comments and answers.
Overall the research questions should be modified and clarified. The presented ones are not adequate because they lack of technological reasons in order to be considered as technological research questions, and they lack of experiments and results if the purpose is to verify that using feedback technology improves vs. using other method (there would be a need for a A / B test). I suggest that the authors modify this throughout the paper to make it consistent.

Author Response
Dear Reviewer thank you so much for taking the time to review our paper one more time. We did some modification to the article based on that and we hope that with this response we are able to clarify some of your concerns.
New Comment:
“Answer: Many thanks for pointing out that this research question can be misunderstood. In order to improve the research question we took various steps: First, we further describe the state of the art of the Learning Hub in section 2.4, Second, On this basis we made the challenge of scalability of real-time data communication explicit. Third, we adjusted the section 3.1 Objectives of the study. Lines 196 - 228. There we first state that the overarching methodology for the research is a design-based research methodology, second we state that the described study is reports on the first iteration of the methodology. There we argue that for this first iteration it is important to explore the technical feasibility of the study and the user experience.
→ OK, this explanation improves the understanding of this part author’s objectives.
We also rephrased the research questions: “RQ1 RQ1 (Technical Feasibility): Is it feasible to send selected sensor data in real-time in order to customize and extend MMLA applications with real-time feedback modules?
→ This sentence defines better what the authors want to test but still is not an application research question but a technological problem. Indeed it is written in the abstract “A technical objective that concerns with the feasibility of extending the PT with an immersive VR module.”
We used this research question to derive a more specific one for our particular use case: RQ1a: Can we use the modular approach facilitated by the LearningHub to extend the PT with VR real-time feedback in order to provide learners with more comprehensive and immersive learning experiences?”
→ To my understanding after reading this part again, this is the objective of the paper and is technological but it is not a research question, and if the authors want to present a technological research question they should present it describing why it is? So far the problem presented only has to do with integration of different components. “
New Answer:
In section 3.1 Lines 234 to 243: We tried to explain in more detail the difficulties of providing immediate feedback. We explained why from the educational perspective real-time feedback is so important. Therefore, we provide an argument for researching how to technically address the problem of real-time feedback.
New Comment:
“→ Table 2 is not properly described, is it related with Appendix A Questionnaire? In that case, why the sentences are different? Please clarify this issue.“
New Answer:
Thanks for pointing out that there could be confusion about it, we changed the questions presented in Table 2 and present the questions that were actually asked, which are the ones of Appendix A.
New Comment:
“Regarding the usability aspect of RQ2, we consider that the way learners experience their interaction with the application will determine the amount of usage of the application, amount of practice of the learners, and therefore how much they learn at the end. If the experience of interacting with it is confusing, boring, or useless, they will never use the application to improve their learning . That is the reason why we consider it important to first do a formative study to explore the user experience before conducting an advanced A-B test experiment to compare the application with traditional learning scenarios. We consider this part of a Design-Based Research methodology that many similar researchers in the field apply for maturing their prototypes.
--> I do agree that is better to go step by step in order to perform the whole research but that doesn’t mean that the final experiments (A/B test using a control group) to test the validity of the study are not included and discussed in order to provide a complete research paper. As the authors say, what is presented is part of it but not all, so, why not to include the whole study? ”
New Answer:
For this first iteration of the overall designed-based research methodology, we decided to start by conducting a formative evaluation (Nierveen & Folmer, 2013). From that article: “. Based on a comparison and synthesis of the definitions of various scholars in the field of formative evaluation (cf. Brinkerhoff, Brethouwer, Hluchyj, & Nowakowski, 1983; Flagg, 1990; Scriven, 1967; Tessmer, 1993), we define formative evaluation in the context of educational design research as a systematically performed activity (including research design, data collection, data analysis, reporting) aiming at quality improvement of a prototypical intervention and its accompanying design principles.”
We consider that the presented study in the paper fits well in the category of formative evaluation, as it allowed us to collect data about the general experience of using the VR module of the PT and how to improve it.
A summative evaluation (A/B testing) definitely will help to confirm and determine the actual effectiveness of the presented added features of the PT. However, conducting an A/B testing for the presented study is unfeasible, we first need to improve the VR-Module based on the results obtained in our study. Then we need to come up with an adequate experimental design and set-up to test the intervention. Some to the steps needed to come up with this experimental design and set-up are:
Identification of the variables that we want to test.
Identification of suitable and comparable interventions for both the treatment and the control group.
The number of interventions.
The duration of the interventions
A study
As stated in (Nierveen & Folmer, 2013):
“It is important not to carry out a summative evaluation until the intervention is developed to such an extent that it has sufficient potential effectiveness. In order to have this potential effectiveness, the intervention should at least be relevant for the educational problem or need at hand, and it should be logically designed and practical in use. This means that design researchers, before entering the summative evaluation phase, need to be able to provide convincing evidence for the quality of the intervention so far, on the basis of the formative evaluation activities undertaken during the development or prototyping phase.”
To avoid confusion we added the following lines to the paper making it explicit that the presented study consists of a formative evaluation:
Abstract
Line14-15 “The described study consists of a formative evaluation and has two main objectives.”
Introduction:
Line 94 “The presented study in this paper consists of a formative evaluation that has two main objectives.”
Section 3.1
Lines 220 - 222 “The study described in this paper is a formative evaluation, which consists of the first full iteration for researching the extension of the PT with the VR module.”
Lines 266-268 “This study presents the first iteration of a design-based research methodology studying the extension of the PT with the VR module. As a formative evaluation, for this iteration, it is important to explore how learners perceive the experience of practicing with this new setup.”
Nieveen, N., & Folmer, E. (2013). Formative evaluation in educational design research. Design Research, 153, 152-169.
Brinkerhoff, R.O., Brethouwer, D.M., Hluchyj, T., & Nowakowski, J.R. (1983). Program evaluation: A practitioner’s guide for trainers and educators. Boston: Kluwer-Nijhof
Flagg, B.N. (1990). Formative evaluation for educational technologies. Hillsdale, NJ: Lawrence Erlbaum Associates
Scriven, M. (1967). The methodology of evaluation. In R.W. Tyler, R.M. Gagné, & M. Scriven (Eds.), Perspectives of curriculum evaluation. AERA Monograph series on curriculum evaluation. nr.1. Chicago, MI: Rand McNally
Tessmer, M. (1993). Planning and conducting formative evaluations: Improving the quality of education and training. London: Kogan Page.
New Comment:
“Line 215-224 these sentences explain very basic communication network knowledge so it should be omitted in this type of paper. A brief subsection describing the whole architecture that has been used would me more useful that describing some parts of it.
Answer: We agree that it can be basic information however we still consider our feasibility research question to be very important. This paragraph for technical people might give no extra value, nonetheless, it allows us to answer our question. Furthermore, we expanded section 2.3 and added Figure 3. to explain the architecture used.
→ This is not a result of the study (or should not be) this is a reflexion of authors about how the study was performed. If you want to keep it should go to summarized at discussion or conclusions sections but not in results. Still the particularities of using some technology should not be a major concern and if they are it may be due a bad decision about what technology has to be used or how to use it.
Reviewer comment: Lines 225-229 doesn’t provide relevant information for the reader. Any practical problem related with the used prototype should be solved before doing the study or it should be done again after they are fixed. Lines 230-236 This lines doesn’t provide insightful information.
Answer: We consider that one of the main purposes for the type of formative study described in this paper is precisely to identify this type of mistakes in real scenarios with small scale studies, before doing a summative evaluation of the system.
→ This is not a result of the study (or should not be) this is a reflexion of authors about how the study was performed. If you want to keep it should go to summarized at discussion or conclusions sections but not in results. Still the particularities of using some technology should not be a major concern and if they are it may be due a bad decision about what technology has to be used or how to use it. ”
New Answer:
This point is connected to the purpose of the presented study, which was a formative evaluation. Under that frame, we consider that the presented results show very important lessons learned on how to get a clear idea of how the application was perceived by users and how to improve its functionality, which is the actual goal of a formative evaluation.
New Comment:
“Answer: Indeed testing the current setup against classical learning techniques is an interesting point to research. However, in the current cycle of the design-based research methodology the study presented in the paper consists of a formative study, testing the effectiveness of different interventions including the current setup of the PT was out of the scope of the study. However, while this is an interesting point that definitely is worth exploring, previous studies have shown that learners do not automatically correct their nonverbal communication by just practicing. Feedback is needed in order to correct some mistakes. The presented study shows already an improvement in the participants performance, showing that the feedback provided by the system was to certain degree effective. We pointed this out in the new version of the paper. Section 5 Discussion lines 433-435. “Concerning learning, the studies in [23,41] show that the mere act of practicing a presentation several times does not lead to measurable improvements, in contrast to practicing a presentation while receiving feedback. ”
→ Ok, still it would be something useful to test for each concrete application and it would be a nice study to have in order to provide a complete research of the work performed.”
New Answer
While we agree that at some point the overarching research should include a summative evaluation with A/B testing to confirm and determine the effectiveness of the proposed application. Conducting and describing a summative evaluation of the presented application is not feasible to be done for this paper, especially considering the current state of the application, and therefore it is outside of the scope of this paper.
Reviewer 4 Report
I am generally OK with the response from the author.
In addition, including a more detailed discussion about the responses to questions 14-18 in the User experience questionnaire will help to improve the paper.
Author Response
Dear reviewer, thanks so much for reading the paper once more and giving us some valuable recommendations.
Comment:
In addition, including a more detailed discussion about the responses to questions 14-18 in the User experience questionnaire will help to improve the paper.
Answer:
We agree that open-ended questions generally can give very interesting insights and are worth to be thoughtfully discussed. We looked once more at the answers provided by the participants to the open-ended questions. Participants generally were not very expressive in their answers and it was difficult to extract more information from them than what we had already reported and discussed. However, added some lines to our paper attempting to expand on the reports and discussion of these responses.
The changes in the paper can be seen in:
Results section
Lines 452- 460
“Regarding the open-ended questions from the questionnaire, when participants were asked: “What, if anything, do you feel like you learned from using the application?”, the most reported answers dealt with the use of pauses, reported by thirteen participants. The next most frequent answer involves the use of gestures, mentioned by four of the participants. These reports align with the objective measurements extracted from the log files of the practice sessions, showing that the area displaying the biggest improvement is the use of pauses, followed by the use of gestures. Concerning gestures, two participants reported being surprised by how their gestures look like. Three participants commented about learning about the feeling of doing a presentation and the importance of practicing them.”
Discussion section
Lines 519-531
In terms of perceived learning, results from the quantitative aspect of the questionnaire indicate that participants perceived to learn a lot while using the PT. Results from the open-ended questions show that participants particularly reported learning about the use of pauses and gestures, which were also the biggest areas of improvement showed by the objective measurements, with an improvement of 33.7% and 32.7% respectively.
Answers to the open-ended questions concerning the most positive aspects of the application show that participants appreciated the ability to practice a presentation while receiving real-time feedback and feeling immersed in a classroom. Consequently, revealing the importance of extending the PT with the VR-module. Moreover, answers to these questions point out the relevance of the tool concerning the development of self-awareness, which the post-practice histogram and video analysis help to obtain. Some participants explicitly pointed out to be surprised about the aspect of their gestures. Overall, results show how in terms of perceived learning the combination of the VR module and post-practice histogram seem to be very helpful.
Round 3
Reviewer 2 Report
Dear authors,
after all these changes the quality of the paper has improved and its clearer to the readers. I still think that an A/B test would improve the scope and quality of the paper as a whole and would be interesting to a broader audience than just technical.
Due to the number of changes there are some new parts that has not been properly introduced such as "selected data" which is an important part of the proposed methodology in order to provide real time. The authors should include the definition of selected data and how is achieved.
Also, a complete English review and proof reading of the paper should be done.
Best,
Author Response
Dear reviewer,
Thanks once more for reviewing the paper. As you recommended we added in Section 2.4 (Lines 255-257) an explanation for all data. In the same Section lines 263-266, we added an explanation for selected data.
Also as recommended we sent our paper to an English proofreader who improved the paper in terms of the English language.
We really appreciate all your suggestions they really helped to improve our paper. We agree with the concerns you raised. Some of them did not fit in this precise study and could not be included. However, be assured that we will consider them for the design of future studies, so thanks so much for your insights.